# 3D Simulation, Electrical Characteristics and Customized Manufacturing Method for a Hemispherical Electrode Detector

**DOI:** 10.3390/s22186835

**Published:** 2022-09-09

**Authors:** Manwen Liu, Wenzheng Cheng, Zheng Li, Zhenyang Zhao, Zhihua Li

**Affiliations:** 1Institute of Microelectronics, Chinese Academy of Sciences, Beijing 100029, China; 2School of Integrated Circuits, University of Chinese Academy of Sciences, Beijing 100049, China; 3School of Physics and Optoelectronic Engineering, Ludong University, Yantai 264025, China; 4School for Optoelectronic Engineering, Zaozhuang University, Zaozhuang 277160, China; 5Shandong Dongyi Photoelectric Instruments Co., Ltd., Yantai 264670, China; 6Shandong Dongyi Institute of Optoelectronic Technology for Industry, Yantai 264670, China

**Keywords:** 3D hemispherical electrode silicon detector, ultra-low capacitance, radiation hardness, full depletion voltage, charge collection efficiency, customized manufacturing method

## Abstract

The theoretical basis of a hypothetical spherical electrode detector was investigated in our previous work. It was found that the proposed detector has very good electrical characteristics, such as greatly reduced full depletion voltage, small capacitance and ultra-fast collection time. However, due to the limitations of current technology, spherical electrode detectors cannot be made. Therefore, in order to use existing CMOS technology to realize the fabrication of the detector, a hemispherical electrode detector is proposed. In this work, 3D modeling and simulation including potential and electric field distribution and hole concentration distribution are carried out using the TCAD simulation tools. In addition, the electrical characteristics, such as I-V, C-V, induced current and charge collection efficiency (CCE) with different radiation fluences, are studied to predict the radiation hardness property of the device. Furthermore, a customized manufacturing method is proposed and simulated with the TCAD-SPROCESS simulation tool. The key is to reasonably set the aspect ratio of the deep trench in the multi-step repetitive process and optimize parameters such as the angle, energy, and dose of ion implantation to realize the connection of the heavily doped region of the near-hemispherical electrode. Finally, the electrical characteristics of the process simulation are compared with the device simulation results to verify its feasibility.

## 1. Introduction

With the development of 3D column [1,2,3,4,5] and 3D trench electrode detectors [6,7,8,9,10,11], it was found that a cylindrical structure has an almost uniform electric field distribution on the cross section [11]. We sought to obtain perfectly uniform electric field distribution by modeling and simulating a 3D spherical electrode detector. In 2021, we proposed and analyzed theoretical basis of a hypothetical spherical electrode detector [12]. However, based on the fabrication technology which exists at present, a spherical electrode detector is only theoretical and hypothetical; can neither fabricate it nor read the output signal. Therefore, to realize such a device, the structure could comprise two identical hemispherical detectors.

The radiation fluence of the upgraded high luminosity Large Hadron Collider (HL-LHC) is predicted to be 1 × 10^16^ 1 MeV n_eq_/cm^2^ in ATLAS and CMS for the innermost layer detectors [13,14]. Recently, 3D electrode detectors have comprised 25% of the ATLAS Insertable B-Layer (IBL) due to their inherent radiation hardness properties [15]. Predictably, 3D electrode detectors have great potential in HL-LHC applications as radiation-resistant detectors [16]. Furthermore, radiation tolerance up to 8 × 10^17^ 1 MeV n_eq_/cm^2^ will be required in future applications, such as the Future Circular Collider (FCC) [17]. For a radiation level of 1 × 10^16^ 1 MeV n_eq_/cm^2^, the trapping problem will be one of the main limiting factors regarding the detector charge collection efficiency (CCE), in addition to the detector full depletion problem.

The trapping distance of free carriers made of silicon decreases after radiation because the silicon lattice suffers damage and defects form upon exposure to high-energy radiation. After 1 × 10^15^ n_eq_/cm^2^ radiation at −20 °C, the effective drift length of electrons decreased to 150 μm, and that of holes dropped to 50 μm. In this case, the detector signal would weaken or even disappear in the plane detector with an electrode spacing of 300 μm to 500 μm [18,19]. Therefore, in this work, the electrode spacing was set at less than 50 μm to provide excellent radiation hardness performance.

In this work, a 3D simulation will be created out using TCAD tools. During the electrical characteristic simulation, the physical models used include Fermi, Doping Dependence, High Field Saturation, Shockley-Read-Hall Generation-Recombination, Auger Recombination, Avalanche, Effective Intrinsic Density, and the Heavy Ion Model. For different particle and photon detection, the incident depth and carrier distribution are different, e.g., the incidence depth of β particle radiation is deep while that of α particles is shallow, and the density of e/h pair is the highest at maximal incidence [20]. In our simulation, the above differences are simply considered as a uniform incidence model, that is, the longitudinal e/h pair is uniformly distributed, and the transverse 1 μm is Gaussian distribution. The degradation of the silicon in detectors is described in [21]. It was noted that this phenomenon results in increased leakage current, bulk resistivity, space charge concentration and free carrier trapping. The space charge concentration is the main factor affecting the detector when the total fluence is in the order of 1 × 10^15^ n_eq_/cm^2^, which increases the full depletion voltage. Free carriers are significantly trapped by radiation-induced defect levels when the total fluence is larger than 1 × 10^16^ n_eq_/cm^2^.

The hemispherical structure of the silicon substrate could be fabricated based on the CMOS and MEMS process. For example, for the 3D hemispherical shell resonators, 3D-SOULE [22], atomic layer deposition (ALD) [23], micro scale glassblowing [24] and other methods were applied. Since the 3D hemispherical electrode detector needs to retain ultra-pure and high resistivity silicon as the detection sensitive bulk, the above process of retaining only the spherical shell is not completely applicable. In addition, a hemispherical electrode is only used to provide good electric field distribution, so the hemispherical quality requirement can be relaxed somewhat. In the current manufacturing process, electrode doping is mostly performed by polysilicon deposition and impurity source diffusion [25,26]. However, the stress caused by polysilicon filling will lead to wafer distortion, resulting in alignment difficulties [27]. In this work, we will investigate the fabrication of a complex 3D near-hemispherical electrode with multi-step deep reactive ion etching (DRIE) [28,29] and deep trench ion implantation using the TCAD-SPROCESS simulation tool. The key is to appropriately set the aspect ratio of the deep trench in the multi-step repetitive process and optimize parameters such as the angle, energy, and dose of ion implantation to realize the connection of the heavily doped region of the near-hemispherical electrode. Finally, the electrical characteristics of the process simulation are compared with the device simulation results to verify its feasibility.

## 2. 3D Modeling and Simulation of a Hemispherical Electrode Detector

The detector schematic is presented in Figure 1 and the structure parameters are as follows: (1) The silicon bulk/substrate is P-type doping with boron, and the concentration is 1 × 10^12^ cm^−3^. The radius of the effective region, namely, the electrode spacing, is 20 μm, which can ensure the radiation hardness of the device. (2) The P+ hemispherical electrode is 1 × 10^19^ cm^−3^ boron doping with 5 μm width, surrounding the whole silicon bulk. A bias voltage is applied to the spherical electrode to make the device work. The P+ hemispherical electrode is defined as the cathode. (3) The collection electrode is in the shape of a hemispherical dot, with a width of 5 μm and 1 × 10^19^ cm^−3^ phosphorus doping. The N+ central dot electrode is defined as the anode. In the hemispherical electrode detector design, the maximum drift distance of the charge carriers and depletion width/depth depend on the electrode spacing rather than the wafer thickness. Compared with the planar pixel detector with electrodes confined to the chip surface, the advantages of the hemispherical electrode detector structure include low capacitance, ultra-fast collection time and low depletion voltage. The output capacitance of the device is lower than those of planar and traditional 3D column electrode detectors due to the dot shape collection electrode. These advantages make the hemispherical electrode detector resistant to radiation-induced increases of depletion voltage and charge carrier trapping in the silicon bulk.

Figure 2 presents 3D and 2D electric potential distributions of the hemispherical electrode detector with different bias voltages. From Figure 2, one can observe that the reverse bias voltage is applied on the cathode, and the potential gradient is relatively uniform throughout the hemisphere. It is worth noting that the values of the potential distribution on the surface of the circular effective silicon region are consistent because of the surface charge density (oxide charge = 4 × 10^11^ cm^−2^) between the silicon and silicon oxide layers, in contrast with oxide charge = 0 cm^−2^. When the oxide charge is 0 cm^−2^, the surface potential has a gradient distribution. 

Figure 3 shows 3D and 2D electric field distributions with a bias voltage range from −5 V to −30 V. The electric field increases with the increasing of bias voltage value. The electric field near the dot electrode is greater than that near the hemispherical electrode in the silicon body because of the PN junction location. However, there is a high electric field region near the hemispherical electrode due to the oxide charge. As shown in Figure 4, the hole concentration value is less than 9.85 × 10^9^ cm^−3^ (the equilibrium carrier concentration of silicon is 1.0 × 10^12^ cm^−3^) when bias voltage reaches 5 V, and the distribution of the carrier concentration does not change significantly with an increase of bias voltage, which means that the device has been fully depleted. In the hemispherical electrode detector, the depletion grows laterally from the dot electrode as bias voltage increases, and the substrate is fully depleted when the depletion region extends fully from the dot electrode to the adjacent central hemispherical electrode.

The minimum voltage required for carriers to reach the saturation drift velocity *v_s_* = 1 × 10^7^ cm/s can be expressed as [30]: (1)$$V_{\min}=\frac{v_{s}}{\mu_{e}}\times R$$

In this work, for *R* = 20 μm and *μ_e_* = 1450 cm^2^/V/s, the minimum voltage *V*_min_ is about 13.79 V. In this situation, the charge collection time along the radius path is *t_c_* = *L_p_/v_s_* = 2 × 10^−10^ s = 200 ps.

## 3. Electrical Characteristics of the Hemispherical Electrode Detector under Radiation

In this section, we simulate and analyze the electrical characteristics of the hemispherical electrode detector in order to study the performance of the device before and after exposure to radiation. The charge density of the saturated oxide layer approaches 3 × 10^12^ cm^−2^~4 × 10^12^ cm^−2^ in the radiation environment [31], resulting in a significant reduction in breakdown voltage. According to the conversion between radiation fluence and oxide charge density, when the radiation fluence is 1 × 10^16^ n_eq_/cm^2^, the oxide charge density is converted to be 2 × 10^12^ cm^−2^ [32,33].

The full depletion voltage for a 3D electrode detector can be estimated as: (2)$$V_{fd}=\frac{qN_{eff}}{6\varepsilon \varepsilon_{0}}(R^{2}-r_{c}^{2})-\frac{qN_{eff}}{3\varepsilon \varepsilon_{0}}R^{2}(\frac{R}{r_{c}}-1)$$
where the elementary charge value *q* = 1.6 × 10^−19^ C, *N_eff_* = 1 × 10^12^ cm^−3^, and the hemispherical electrode radius *R* = 20 μm. The product of the silicon dielectric constant and the vacuum dielectric constant εε_0_ = 11.9 × 8.854 × 10^−14^ F/cm. Based on this calculation, the full depletion voltage of the hemispherical electrode detector without radiation is only −3.75 V. In addition, the full depletion voltage can be extracted from the simulation data in Figure 5a (the abscissa corresponds to the point of leakage current saturation); the simulated value for depletion voltage before irradiation is basically the same as the theoretical calculation value. Additionally, the full depletion voltage increases with increasing radiation fluence. The full depletion voltage is much lower than that in a planar detector, since it is not dependent on the silicon wafer thickness, but rather, only the electrode spacing. 

From Figure 5a, we can observe that when the leakage current increases with the radiation fluence, the latter increases and the growth trend becomes more obvious. Figure 5b shows the junction breakdown characteristics of the detector. The abscissa of the point where the dark current suddenly increases is the junction breakdown voltage at this fluence. The breakdown voltage is above 200 V without radiation. However, the overall breakdown voltage will be dominated by the surface breakdown voltage when the surface charge density is larger than 4 × 10^11^ cm^−2^, such as 3 × 10^12^ cm^−2^. Therefore, the simulation results can be used as a reference for controlling the variable of surface charge density in the manufacturing process.

Figure 6 presents the geometry capacitance of the hemispherical electrode detector with different collection electrode radii. The capacitance ranges from 0.8 fF to 2 fF when the radius ranges from 1 μm to 2.5 μm; the larger the collecting electrode radius, the larger the capacitance.

As charged particles pass through the detector, inelastic collisions occur through Coulomb forces interacting with atomic electrons. The Bethe-Block formula describes the average energy loss per unit path of a charged particle:(3)$$-\frac{dE}{dx}=4\pi N_{A}r_{e}^{2}m_{e}c^{2}z^{2}\frac{Z}{A}\frac{1}{\beta^{2}}\left[\frac{1}{2}\ln(\frac{2m_{{}^{2}}c^{2}\beta^{2}\gamma^{2}T_{\max}}{I^{2}}-\beta^{2}-\frac{\delta (\gamma )}{2})\right]$$
where *N_A_* is Avogadro’s number, *r_e_* is the classical electron radius, *m_e_* is the electron mass, *c* is the speed of light, *z* is the charge of the incident particle, *Z* is the atomic number, and *A* is the atomic mass of the considered material. *β = v*/*c* represents the velocity, *T*_max_ the maximum kinetic energy that can be imparted to a free electron in a single collision, *I* the mean ionization energy, and *δ* the density effect correction. The energy of a particle must be higher than the −*dE*/*dx* minimum to produce a significant signal, which is called the minimum ionizing particle (MIP). The MIP is commonly used to evaluate detector performance. In this paper, the performance of detectors with different incident positions and incident depths is studied, as shown in Figure 7.

The MIP simulation utilizes the heavy ion incidence model. The distribution of electron-hole pairs produced by different kinds of energetic particles is different. In this paper, a uniform distribution setting is adopted, with uniform distribution in the longitudinal direction and Gaussian distribution in the transverse direction. We assume that 80 electron/hole (e/h) pairs can be produced by MIP-like particles. The *LET_F* is set as 1.28 × 10^−5^, and the *wt_hi* is 1 μm.

Figure 8 presents the I-t curves of the hemispherical electrode detector with different radiation fluences; (a) incident position *R* = 10 μm, and the incident depth is 5 μm; (b) incident position *R* = 10 μm, and the incident depth is 10 μm. For the same radiation fluence and incident position, the induced current increases with an increase of the incident depth. The leakage current increases with the radiation fluence, but the induced current decreases, as does the full width at half maximum (FWHM) of the induced current. The FWHM of the induced current vs. time curve is 1.424 × 10^−10^ s without radiation. Figure 9 shows the charge collection efficiency (CCE) curves with different particle incident positions. The charge collection efficiency decreases as the radiation fluence increases. When the fluence is ≤ 1 × 10^16^ n_eq_/cm^2^, the CCE decreases slowly, which indicates that the 3D detector has strong radiation resistance. The CCE is larger than 90% when the radiation fluence is less than 1 × 10^16^ n_eq_/cm^2^, no matter where the incident position is. When the fluence increases further, the CCE decreases rapidly.

## 4. A Customized Manufacturing Method of a Hemispherical Electrode Detector

The manufacturing process of a 3D hemispherical electrode silicon detector, combined with the process simulation, are presented in this section. In the process simulation, due to the highly symmetrical nature of the hemispherical structure, the process simulation of the 2D section, as shown in Figure 10a, can fully reflect the process of the 3D structure. The detector is fabricated on a high resistivity, boron-doped, P-type (100) single-crystal silicon (SCS) substrate. In the practical process, a SOI or silicon–silicon direct wafer bonding (Si–Si DWB) substrate can be used to avoid wafer cracking [34]. Different aspect ratios are used for etching according to the spherical change rate. The trench depth and width corresponding to aspect ratio at different *x* positions is presented as follows:(4)$$h(x)=H-\sqrt{R^{2}-x^{2}}-l\;\;(x\geq 0)$$
(5)w(x)=kh(x)
where *H* is the thickness of the wafer, *R* is the radius of the hemispherical electrode, *l* is the junction depth, *h* is the deep etching depth, *w* is the deep etching width, and *k* is the reciprocal of the depth width ratio. For example, if the change rate of *h_1_* to *h_2_* in Figure 10b is fast, an aspect ratio of 30:1 is used. When etching the center of the spherical surface (*x* = 0), an aspect ratio of 10:1 can be used. 

Figure 11 shows the fabrication process of the hemispherical/near-hemispherical electrode detector:A 2-μm silicon oxide layer is grown on a silicon surface as a hard mask for the subsequent DRIE and boron ion implantation process. The first anisotropic etching was performed using the Bosch process (shown as Figure 11a).Ion implantation of the deep trench. The simulation result shows that an implantation angle of 7° and a wafer rotation of 7° can make the bottom of the deep trench and the side wall uniformly doped. In practice, only the bottom of the trench and the bottom of the trench sidewall need to be doped (Figure 11b).The surface silicon oxide layer is removed (Figure 11c).The oxide layer is deposited on the silicon surface and in the trench again, and the anisotropic, 2-μm oxide deposition is set directly in the simulation (Figure 11d).The above steps are repeated to create a near-hemispherical electrode (Figure 11e–g).Finally, the central electrode and metal layers for the power supply circuit and readout circuit are fabricated by a CMOS process on the back of the wafer (Figure 11h).

It should be noted that due to the limitation of the simulation model and considering the depth of the required calculations, details such as oxide deposition and etching need to be adjusted according to the practical situation in future experiments. The simulation processes and results are shown in Figure 12 and Figure 13. The first deep trench etching of spherical electrode is located at *x* = R on the front of the wafer, and the deep trench ion implantation depth may not reach *x* = R on the back. Therefore, another ion implantation is also required on the back to connect the P+ electrode. 

The change rate of *h(x)* is fast near the hemisphere *x* = R. In order to ensure the near spherical structure, the difference between the first two deep trench etching depths *h_1_* and *h_2_* is the largest. After high temperature annealing, it is difficult to connect the heavily doped region of the near hemisphere electrode. Therefore, it is necessary to accurately set the energy, dose, implantation angle and wafer rotation angle of the deep trench ion implantation. The simulation shows that an implantation tilt of 7° and a wafer rotation of 7° can make the doping of the deep trench bottom connect with that of the left wall. The doping concentration and width of the left wall are the keys to achieving interconnections among the boron ions which are implanted in each step. 

Here, a brief explanation of the deep trench ion implantation simulation must be provided. In this paper, we describe the simulation of the preparation of hemispherical electrodes by ion implantation. By controlling the parameters such as implantation energy and dose, the doping distribution of deep trenches can be controlled more directly and accurately, which is critical for the connection of the hemispherical electrodes. However, due to the limitation of computing power, the above parameters are based on 2D simulation. In the actual process, the ion implantation tilt angle and the wafer rotation angle will greatly affect the doping situation, and as such, a 2D simulation cannot fully correspond to the actual process. Therefore, in future, we will do further research on the process parameters for the ion implantation of the annular and columnar trenches in the laboratory. At present, when constructing 3D detectors, doping in deep trenches is mainly achieved by solid-state source diffusion; hemispherical electrodes can also be prepared by this method. Filling deep trenches with polysilicon will introduce stress and cause wafer bending. Although solid-state source diffusion may be less able to control the dopant concentration profile than ion implantation, its preparation technology is relatively mature, and it is also a good method which is worth trying here.

Figure 14a–c show the doping profiles after annealing with different diffuse times. With increasing diffuse time, the number of doping regions in the bottom and side wall increases. Based on the ion implantation angle simulation parameter settings applied in this paper, Figure 15 shows doping curves with different implantation energies and diffusion time. The doping dose is 1 × 10^15^ cm^−2^, the implantation energies are 80 KeV and 360 KeV, and the diffusion times are 1, 30, and 60 min at 1050 °C. The activated left wall Boron concentration distribution has a width of 0.15 μm to 0.5 μm and a peak concentration range of 2.85 × 10^17^ cm^−3^ to 7.80 × 10^17^ cm^−3^; the higher the ion implantation energy, the deeper the boron ions are able to penetrate the silicon and the lower the peak concentration. Sidewall doping is used as the “wire” of the hemispherical electrode; therefore, in the actual process, it is necessary to carry out detailed experimental research on various ion implantation angles to ensure the conduction of the hemispherical electrode. The peak concentration of the sidewall doping (17 orders of magnitude) is lower than that of the trench bottom doping (19 orders of magnitude); this is due to the small tilt angle and the fact that a large number of ions are implanted into the bottom of the trench.

Based on the process steps shown in Figure 11, we now construct a 2D cross-section model of the device and demonstrate its electric characteristics, as shown in Figure 16. The reverse bias voltage is −15 V. In Figure 16, the potential distribution, electric field and the e current density profiles are presented. It can be seen that in the effective area of the device, the electric characteristics generated by the process simulation are basically consistent with those of the simulation results, which indicates that the construction process is valid.

Figure 17 shows a comparison of devices with different electrode structures. Figure 18 presents a comparison of the leakage currents of different device structures, i.e., with wafer thicknesses of 30 μm and 300 μm. In Figure 18a, the black and green curves indicate the leakage current results of the hemisphere electrode detector from device simulation and the process simulation, respectively. One can observe that they coincide, which further indicates that the proposed fabrication process is valid. The leakage current in 3D trench type structures is larger than that of the 3D hemisphere electrode detector; this trend becomes increasingly obvious when the wafer thickness increases.

## 5. Conclusions

A 3D model of a hemispherical electrode detector has been built in this work. The electrode spacing was set as 20 μm, i.e., less than the trapping distance of free carriers in silicon for a radiation fluence less than 1 × 10^16^ n_eq_/cm^2^ to avoid the trapping problem, decrease the full depletion voltage and improve the response time and CCE. Additionally, 3D simulations of potential, electric field and carrier concentration distributions were carried out. 

Furthermore, we simulated the I-V, C-V characteristics, induced current and CCE of the device, and found that the device had the characteristics of low dark current, small capacitance, low depletion voltage and good radiation resistance. According to our calculations, before exposure to radiation, the full depletion voltage was below 1V. The breakdown voltage was above 200 V without radiation. The capacitance ranged from 0.8 fF to 2 fF when the radius of collection electrode ranged from 1 μm to 2.5 μm. For the same radiation fluence and MIP-like particle incident position, the induced current increased with increasing incident depth. The leakage current increased while the induced current decreased with the radiation fluence. The FWHM of the induced current vs. time curve was 1.424 × 10^−10^ s without radiation. The CCE decreased as the radiation fluence increased. The CCE was larger than 90% when the radiation fluence was less than 1 × 10^16^ n_eq_/cm^2^, regardless of where the incident position was.

A customized fabrication method for a complex, 3D, near-hemispherical electrode with multi-step deep reactive ion etching (DRIE) and deep trench ion implantation is proposed. The process simulation is presented in this work. The main objectives were to set an appropriate aspect ratio for deep trench etching in a multi-step, repetitive process and to optimize parameters such as the angle, energy, and dose of ion implantation to ensure the connection of the heavily doped region of the near-hemispherical electrode. The activated left wall boron concentration distribution had a width of 0.15 μm to 0.5 μm and a peak concentration of 2.85 × 10^17^ cm^−3^ to 7.80 × 10^17^ cm^−3^. Finally, the electrical characteristics, as determined by a simulation, indicated that the manufacturing process method is valid. As such, the present research may provide a reference for future process experiments. The leakage current of the 3D hemisphere electrode detector was less than that of a 3D trench structure; this trend was most obvious when the wafer thickness was greater.

## Figures and Tables

**Figure 1 sensors-22-06835-f001:**
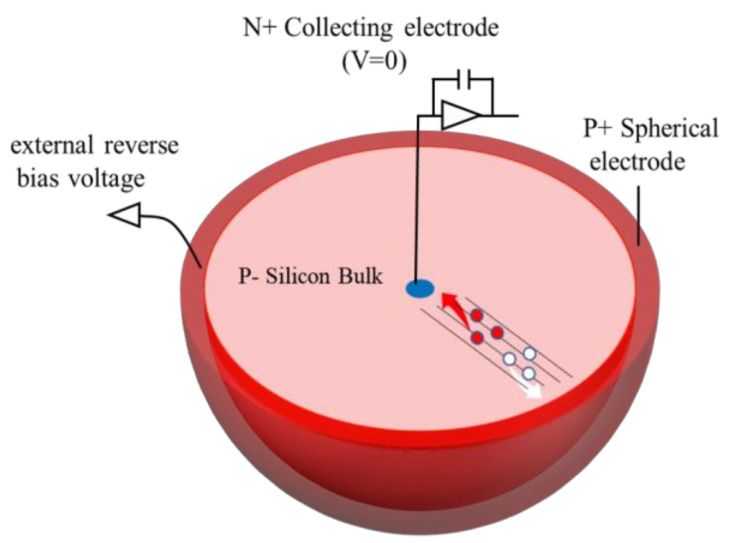
Schematic of the hypothetical hemispherical electrode detector.

**Figure 2 sensors-22-06835-f002:**
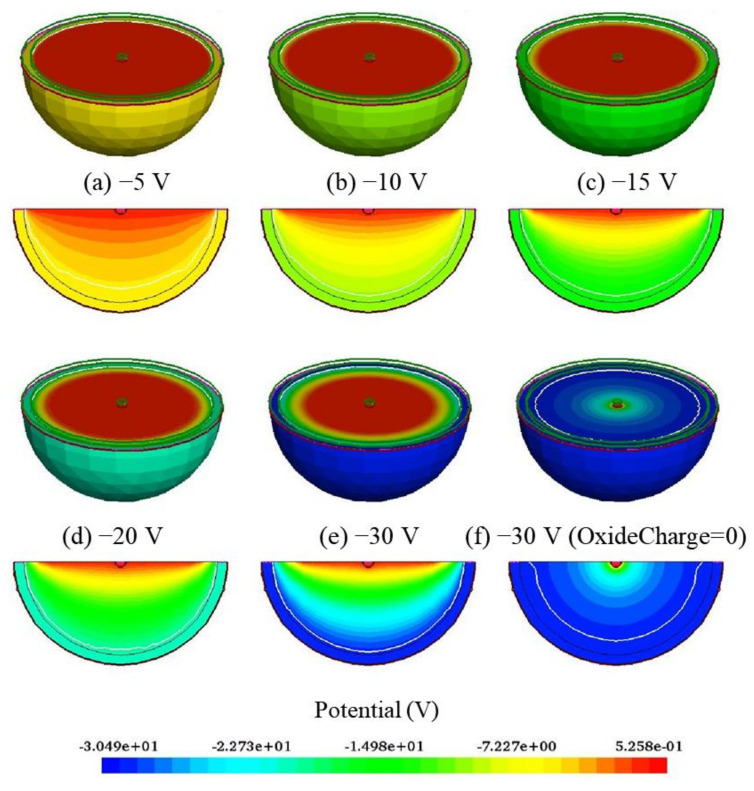
3D and 2D electric potential distributions with different bias voltages.

**Figure 3 sensors-22-06835-f003:**
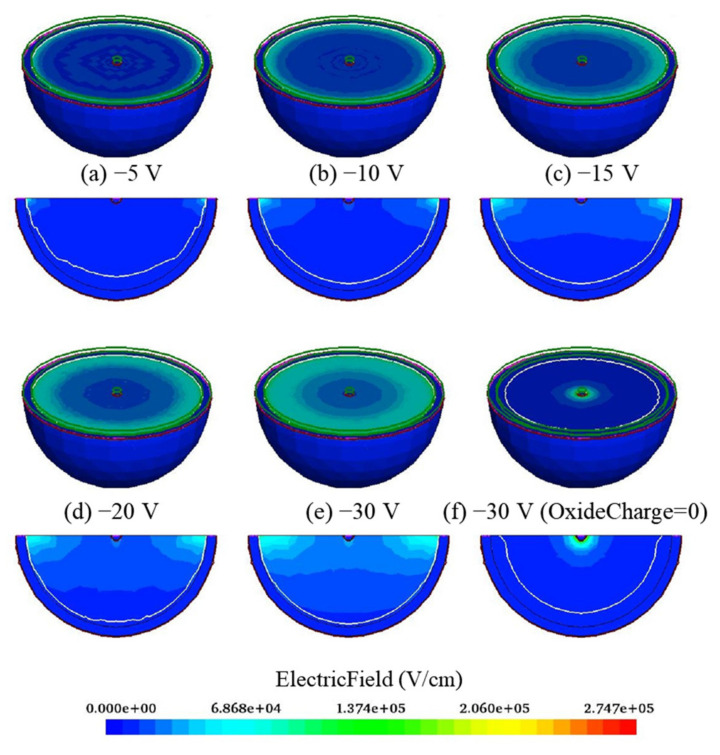
3D and 2D electric field distributions with different bias voltages.

**Figure 4 sensors-22-06835-f004:**
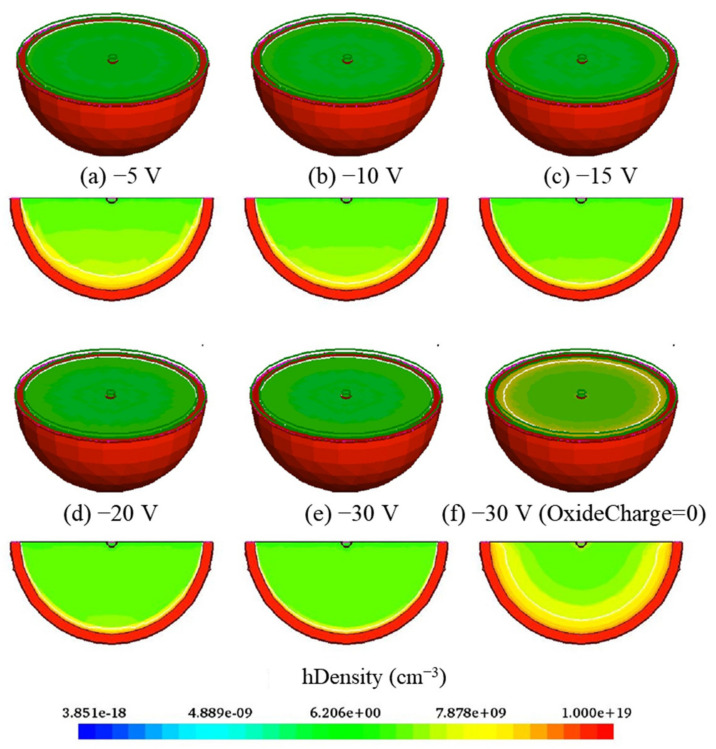
3D and 2D hole concentration distributions with different bias voltages.

**Figure 5 sensors-22-06835-f005:**
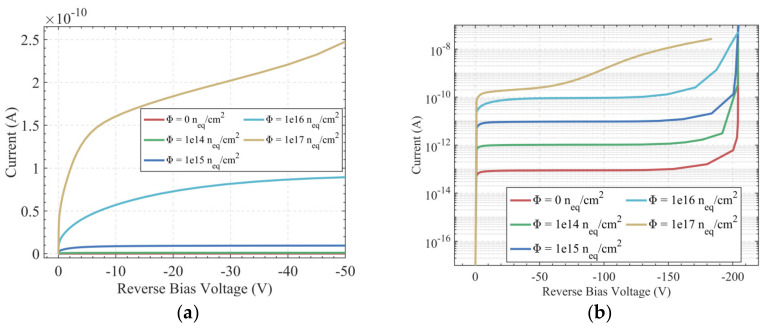
Leakage current of the hemispherical electrode detector; (**a**) Leakage current with bias voltage of −50 V; (**b**) Leakage current with bias voltage of −200 V.

**Figure 6 sensors-22-06835-f006:**
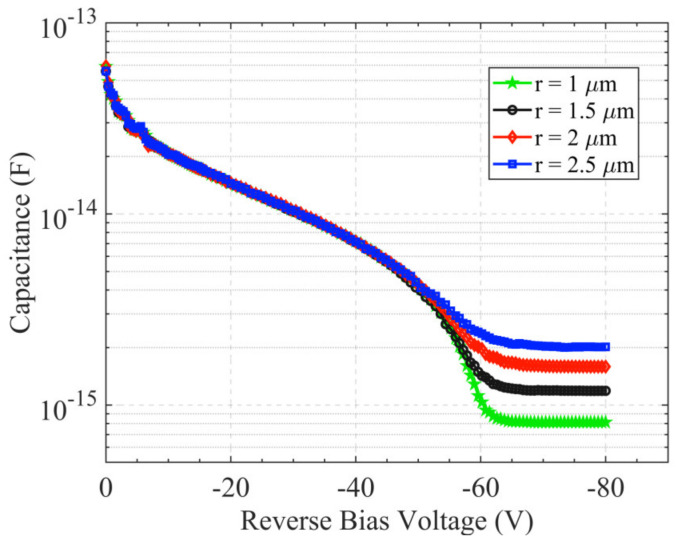
Capacitance of the hemispherical electrode detector.

**Figure 7 sensors-22-06835-f007:**
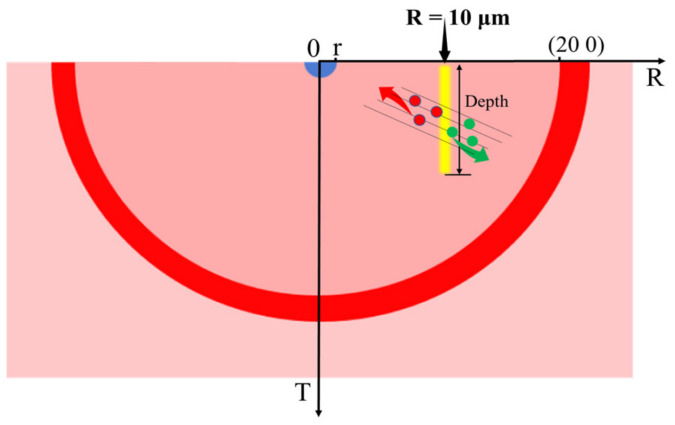
Schematic diagram of minimum ionizing particle (MIP) incidence and unbalanced carrier drift path.

**Figure 8 sensors-22-06835-f008:**
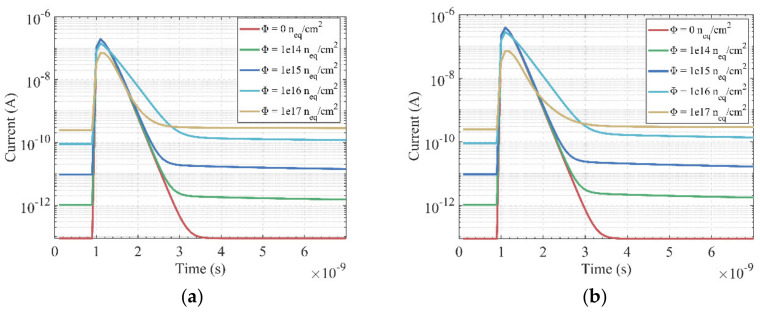
I-t curves with varying radiation fluences; (**a**) particle incident position *R* = 10 μm and the incident depth is 5 μm; (**b**) incident position *R* = 10 μm and the incident depth is 10 μm.

**Figure 9 sensors-22-06835-f009:**
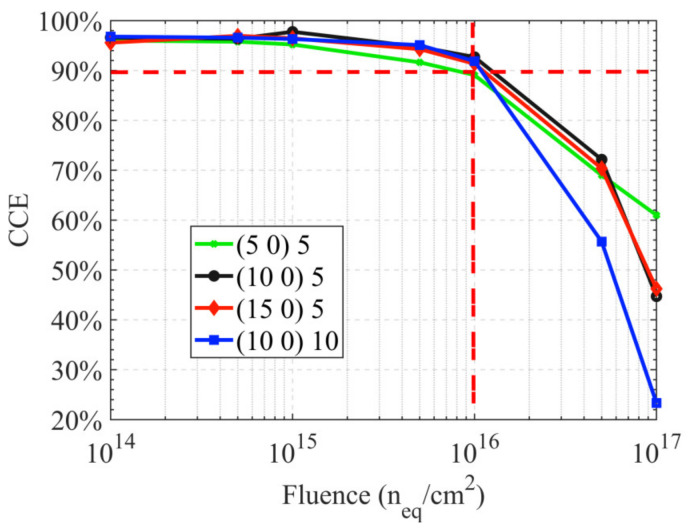
The charge collection efficiency (CCE) curves.

**Figure 10 sensors-22-06835-f010:**
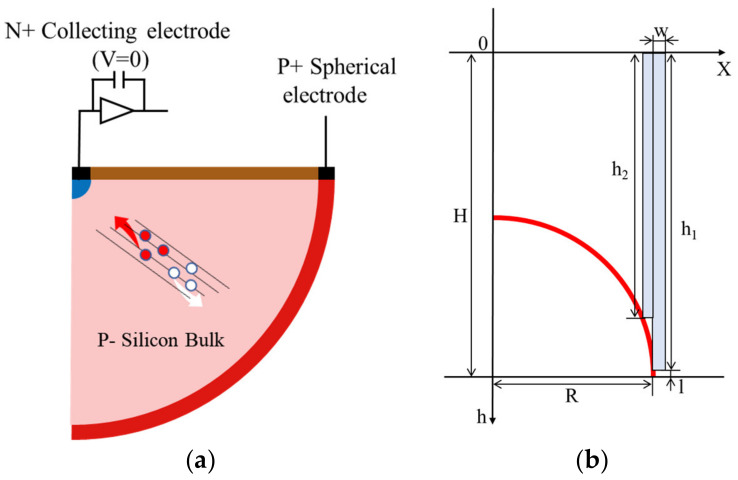
(**a**) 2D profile of the hemispherical electrode detector for process simulation; (**b**) The process simulation structure parameters.

**Figure 11 sensors-22-06835-f011:**
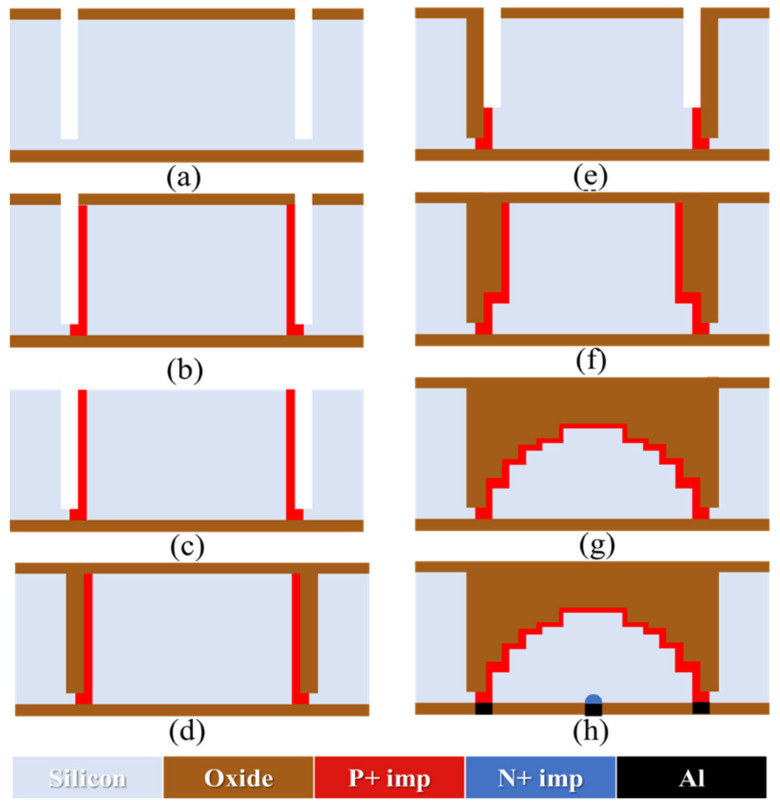
The proposed fabrication process of the hemispherical/near-hemispherical electrode detector.

**Figure 12 sensors-22-06835-f012:**
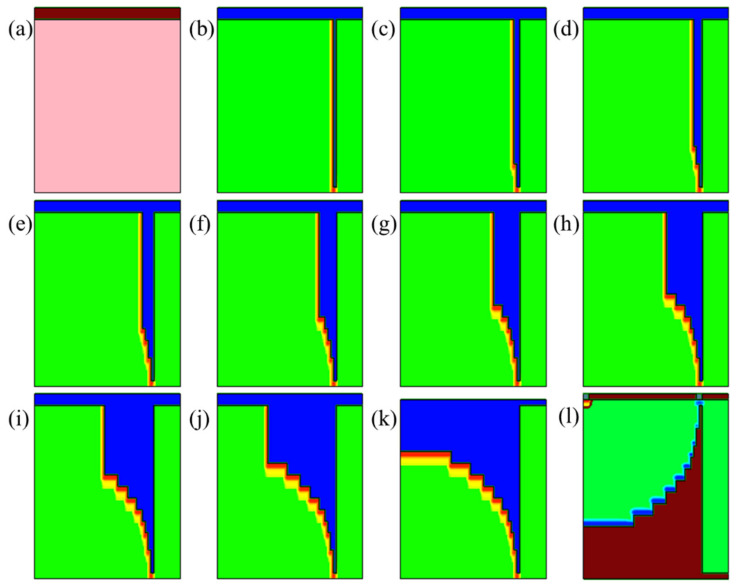
The fabrication process simulation of the hemispherical/near-hemispherical electrode detector (**a**–**l**).

**Figure 13 sensors-22-06835-f013:**
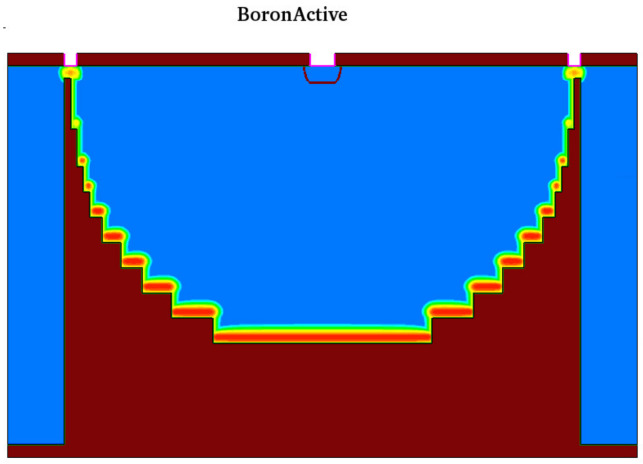
The simulation result of the hemispherical/near-hemispherical electrode detector.

**Figure 14 sensors-22-06835-f014:**
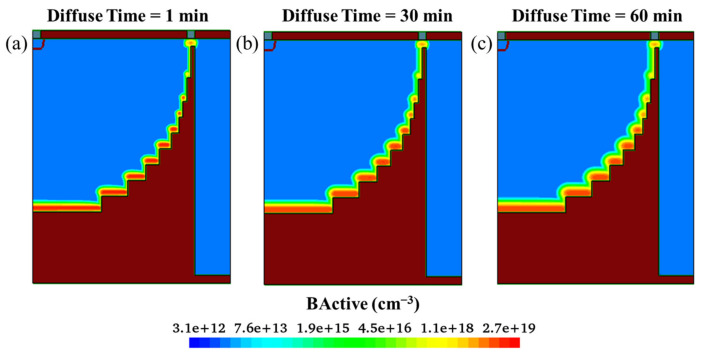
Doping profiles after annealing with different diffuse times (**a**) 1 min; (**b**) 30 min; (**c**) 60 min.

**Figure 15 sensors-22-06835-f015:**
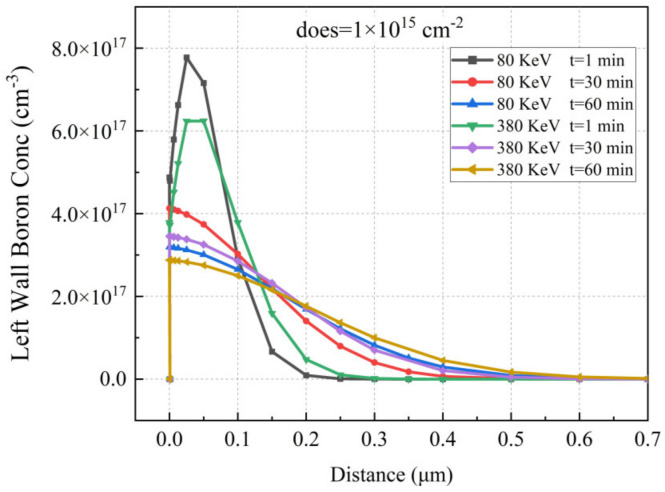
Comparison of doping curves with different implantation energy and diffusion times.

**Figure 16 sensors-22-06835-f016:**
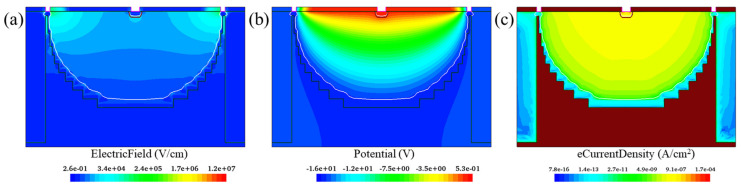
(**a**) Electric field distribution; (**b**) Electric potential distribution; (**c**) E current density distribution.

**Figure 17 sensors-22-06835-f017:**
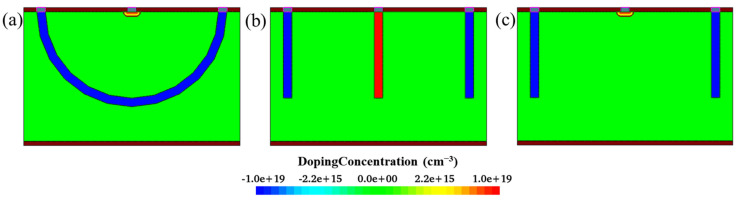
Comparison of different device structures; (**a**) The hemispherical electrode detector; (**b**) 3D trench detector with column collecting electrode; (**c**) 3D trench detector with dot collecting electrode.

**Figure 18 sensors-22-06835-f018:**
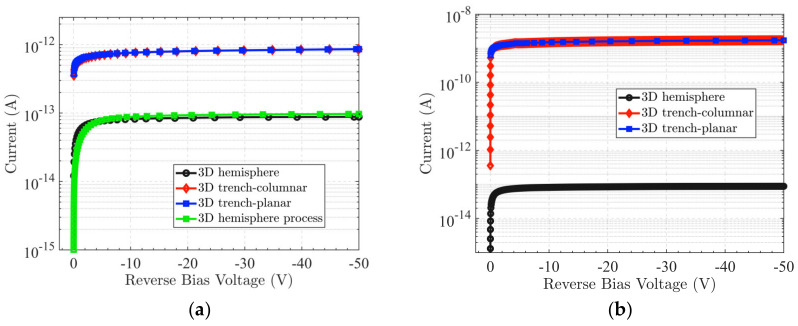
Leakage current comparison of different device structures; (**a**) Wafer thickness of 30 μm; (**b**) Wafer thickness of 300 μm.

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
