# Peer review of "3D Simulation, Electrical Characteristics and Customized Manufacturing Method for a Hemispherical Electrode Detector"

_sensors, 2022, doi:10.3390/s22186835_

Round 1

Reviewer 1 Report

The paper presents development of a novel hemispherical electrode silicon detector, including the TCAD simulation and a possible fabrication method. The simulation setup and results were well described. The paper is in good form and ready for publication.

Author Response

Thanks for the reviewer’s comment.

Reviewer 2 Report

This manuscript reported an interesting study focusing on the simulation of hemispherical electrode detector. The paper is well organized with proper data support. I would suggest a detailed comparison between this work and other reported literature (maybe both simulation and experimental) for better demonstration and highlighting the significance of this work. In addition, the English writing can be further improved.

Author Response

Thanks for the reviewer’s comment. The details are presented in the cover letter.

Reviewer 3 Report

This is an interesting work on modelling and simulation of the production/fabrication of a hemispherical electrode detector using well known CMOS technology. The 3D model seems to be sound and sufficiently accurate to be able to give interesting results. The simulation of the behaviour of the hemispherical electrode detector model is well done and even if limited to the fundamental I-V and capacitance-voltage characteristics, induced current and CCE the results prove at a reasonable level the interest of the approach. A multistep deep reactive ion etching and deep trench ion implantation is also proposed for the fabrication of 3D near-hemispherical electrode.

It is not interely clearly justified the statement abot the activated left wall Boron concentration for all conditions on line 272 to 275 and figure 15. Although being a modelling and simulation work, and valid as such, we advise authors to briefly discurse on the implementation problems and difficulties, and potential of the proposed method. Certainety authors will in the future explore other options for the design of the method. The paper will benefit if authors briefly talk about those prospects.

Reviewer 4 Report

Reference: sensors-1877935

Review Report

The MS entitled “3D simulation, electrical characteristics and customized manufacturing method study of the hemispherical electrode detector” by Manwen Liu, Wenzheng Cheng, Zheng Li, Zhenyang Zhao and Zhihua Li reports on a 3D TCAD study of a CMOS compatible detector with a hemispherical electrode. The study is realized using commercial software and conventional models. A discussion of the possible fabrication of such device is given.

The English of the paper is relatively good and the paper is well organized. Nevertheless, I advise authors about the need for a careful check of the MS language as, for, instance on page #4 line #129 it was written “ …due to the exist of the surface charge density”.

I think the paper can be considered for publication one the next few issues have been addressed:

1.       MIP should be defined (the acronym is only explained in Figure 7) and the concept discussed in the text

2.       As the work is only based on TCAD authors must give the parameters used in each model.

3.       May authors look for experimental results to compare with their simulation?

4.       I understand that “avalanche” means that “impact ionization” is considered in the work. Is that true? I think that electric fields as intense as 150kV/cm have been considered, in this case avalanche (at least an onset) may take place: could authors give details about the level of avalanche in the presented results? Authors discussed about this on page #4 but they do not presented the TCAD results.

5.       May authors give details about the Heavy Ion Model?

Authors wrote about creation of defects due to radiation but I do understand where and how this was tak

Round 2

Reviewer 4 Report

Authors have responded satisfactorily to the question on my first report.

I will recommend the MS for publication.